# Comparison of Ultrasound Image Classifier Deep Learning Algorithms for Shrapnel Detection

**DOI:** 10.3390/jimaging8050140

**Published:** 2022-05-20

**Authors:** Emily N. Boice, Sofia I. Hernandez-Torres, Eric J. Snider

**Affiliations:** Engineering, Technology, and Automation Combat Casualty Care Research Team, United States Army Institute of Surgical Research, Ft. Sam Houston, TX 78234, USA; emily.n.boice.ctr@mail.mil (E.N.B.); sofia.i.hernandeztorres.ctr@mail.mil (S.I.H.-T.)

**Keywords:** deep learning, ultrasound imaging, image interpretation, artificial intelligence, shrapnel, military medicine, emergency medicine

## Abstract

Ultrasound imaging is essential in emergency medicine and combat casualty care, oftentimes used as a critical triage tool. However, identifying injuries, such as shrapnel embedded in tissue or a pneumothorax, can be challenging without extensive ultrasonography training, which may not be available in prolonged field care or emergency medicine scenarios. Artificial intelligence can simplify this by automating image interpretation but only if it can be deployed for use in real time. We previously developed a deep learning neural network model specifically designed to identify shrapnel in ultrasound images, termed ShrapML. Here, we expand on that work to further optimize the model and compare its performance to that of conventional models trained on the ImageNet database, such as ResNet50. Through Bayesian optimization, the model’s parameters were further refined, resulting in an F1 score of 0.98. We compared the proposed model to four conventional models: DarkNet-19, GoogleNet, MobileNetv2, and SqueezeNet which were down-selected based on speed and testing accuracy. Although MobileNetv2 achieved a higher accuracy than ShrapML, there was a tradeoff between accuracy and speed, with ShrapML being 10× faster than MobileNetv2. In conclusion, real-time deployment of algorithms such as ShrapML can reduce the cognitive load for medical providers in high-stress emergency or miliary medicine scenarios.

## 1. Introduction

Ultrasound (US) imaging is commonly used in medicine for its nondestructive testing capabilities and real-time assessment value. One such example is the detection of foreign bodies during emergency medicine assessments due to its high accuracy, instrument portability, and modest power requirements [1,2,3]. Higher-resolution imaging modalities (CT, MRI, etc.) are preferred for diagnosis in hospital settings, but this is typically not possible in remote settings, such as combat casualty care. In addition, acquisition and interpretation of US images can only be effective if the end user is trained in sonography and anatomy. This can be a technically challenging process, requiring hours of training.

Algorithms for ultrasound imaging diagnostics were developed for a range of use cases, such as detecting tumors [4], thyroid nodules [5], and lung pathologies in COVID-19 patients [6]. These types of algorithms primarily rely on supervised deep learning convolutional neural networks (CNNs) to identify trends in image sets. More advanced algorithms can utilize object detection or segmentation approaches to highlight precise regions in the US image field as abnormal [7] or be used in real time [8,9,10]. The use of artificial-intelligence-guided diagnostics would enable faster, higher accuracy assessments, which could be critical in resource- and personnel-limited, high-stress environments, such as battlefield trauma scenarios.

We previously developed [11,12] and tested a deep learning model (ShrapML) for the automated detection of shrapnel. ShrapML is an image classification model trained on ultrasound images of both gelatin tissue phantoms and porcine tissue. When tested with ultrasound images for external validation, ShrapML makes binary predictions as to whether shrapnel is present in the image, with an overall accuracy of 95%, an F1 score of 0.95, and an area under the receiver operating characteristic (ROC) curve of 0.95. 

Other existing algorithms have conventionally been trained with the ImageNet dataset [13] to determine thousands of category types. These algorithms can be computationally intensive and require millions of images for training. Such algorithms include VGG16 [14], EfficientNet [15], and InceptionNet [16,17,18], among others. Although the ImageNet dataset is extensive, it is focused on everyday objects that are not relevant to medical imaging. Through transfer learning, these models can be used on new medical imaging datasets and compared for accuracy. As high accuracy with lower computational power needs will be ideal for incorporating image analysis algorithms into existing ultrasound hardware, here, we enhance the ShrapML classifier using Bayesian optimization and then compare its performance to that of conventional, well-established image classifier algorithms using an expanded US shrapnel dataset. 

## 2. Materials and Methods

CNN models can achieve human-like accuracies in image classification problems due to their self-learning and superior classification abilities. A CNN network is typically comprised of a chain of organized components: convolution layers (Conv) with activation functions, max pooling layers (Pool), and batch normalization operations. The hierarchical network structure provides high-level feature maps, reduced computational complexity, and improved generalization ability. 

### 2.1. ShrapML Architecture

Previous work from the laboratory outlines the full architecture of the ShrapML classifier algorithm [11]. In short, the algorithm was built using TensorFlow/Keras libraries and Jupyter Notebook in Python. Red-green-blue (RGB) ultrasound images were used as an input for this algorithm and resized. Images were augmented by random flip, rotation, zoom, and contrast processes, and this dataset was used to train the model. The model consisted of a series of 5 repeated successions of Conv with ReLu activators and Pool layers with increasing filter size. This was followed by a dropout layer, a flatten layer, and, finally, a dense layer with sigmoid activation. This model was trained over 100 epochs using RMSprop optimizer to minimize validation loss. When testing this model with externally validated images, the model predicted whether an image was positive or negative for shrapnel and gave a confidence value of its prediction. 

### 2.2. Bayesian Optimzation Setup

For optimization of the hyperparameters in ShrapML, beyond the initial iterative approach, we used Bayesian optimization. This is a widely used approach to remove human bias from the model optimization aspect and evaluate hyperparameters with a probabilistic approach [19,20,21,22]. ShrapML was set up to optimize 5 hyperparameters of the model (Table 1). The CNN size and layers were set up such that the size of the next layer was a product of the CNN size and layer number for each additional layer (4, 8, 12, or 16 filters for a 4-layer model, for instance). The phantom image sets used to originally develop ShrapML were used as training and validation datasets during optimization. A total of 10 epochs were performed for each iteration, with the goal of optimizing the problem set to minimize validation loss. Bayesian optimization was set up using the experiment manager application in MATLAB R2021b (Mathworks, Natick, MA, USA) and was concluded after 250 trials were evaluated. Optimization was performed on an HP workstation (Hewlett-Packard, Palo Alto, CA, USA) running Windows 10 Pro (Microsoft, Redmond, WA, USA) and an Intel Xeon W-2123 (3.6 GHz, 4 core, Santa Clara, CA, USA) processor with 64 GB RAM. The top five models were compared for training, validation accuracy, and loss, as well as the total training time through the 10 epochs, to select the optimized algorithm architecture.

### 2.3. Ultrasound Phantom Development

A previously published gelatin tissue phantom [12] was modified for use in this work. In short, a 3D-printed mold was designed with human adult male thigh dimensions, including three major components: a 3D-printed bone, a muscle layer, and a fat layer. For the construction of the muscle and fat layers, a 10% (*w*/*v*) gelatin (Thermo-Fisher, Waltham, MA, USA) solution was prepared using a 2:1 solution of water and evaporated milk (Kirkland, Seattle, WA, USA). The fat-layer gelatin solution was supplemented with 0.1% flour (HEB, San Antonio, TX, USA) for increased hyperechoicity. The muscle-layer gelatin solution was supplemented with 0.25% flour and roughly chopped pieces of 2% agarose (Sigma-Millipore, St. Louis, MO, USA) for added heterogenicity. The mold was assembled first for the inner layer (muscle layer). The inner layer solution was poured and inverted repeatedly to keep the agarose pieces distributed as the phantom solidified. After approximately 30 min, the inner layer stiffened and was placed in the outer-layer mold. The second layer (outer fat layer) was poured around the inner layer and chilled at 4 °C. After solidification, the completed tissue phantom was removed from the mold and used for US imaging applications.

### 2.4. Ultrasound Image Collection and Processing

After the tissue phantom was created, ultrasound images were collected using a Sonosite Edge (Fujifilm Sonosite, Bothell, WA, USA) and HFL50 ultrasound transducer (Fujifilm Sonosite, Bothell, WA, USA). To avoid air interference, all phantom imaging was performed underwater. Baseline data were collected prior to shrapnel insertion of the entire phantom using 10 s B-mode clips. For shrapnel, we previously determined that ShrapML performed similarly with different material types, so a single material type of varying length was used for this study [11]. A 2.5 mm diameter brass rod was cut to 2, 4, 6, 8, or 10 mm length fragments to provide a range of shrapnel sizes (Figure 1). The brass pieces were then inserted to varying depths within the phantom in the four quadrants using surgical forceps, and shrapnel-positive data were collected with 10 s B-mode clips.

Ultrasound video clips were transferred from the imaging device, and frames were extracted from the clips using the ffmpeg-Ruby tool (version 4.4). Duplicate frames were removed from the dataset, as the US clips had a high frames-per-second rate, resulting in four duplicates for every new frame. Individual images where then sorted into ground truth groups: shrapnel (positive) and baseline (negative). Any image for which it was uncertain as to which group it belonged to was discarded and not used in the training dataset. Next, images were cropped and standardized to remove the settings, file name, and miscellaneous US information included in the exported frames, followed by conversion to 16 bit and resizing to 512 × 512. All image processing operations were performed using the batch image processing toolkit in MATLAB R2021b. In total, three different phantoms were imaged, resulting in approximately 6600 baseline and 6700 shrapnel images.

### 2.5. Model Training Overview

Model training was performed for three separate experiments detailed below. Specific differences for each experiment are detailed in each subsection. First, shrapnel and baseline images were subsplit into three groups: 60% training, 20% validation, and 20% testing. Training images were augmented in an attempt to prevent overfitting by applying up to ± 20% zoom to each image, ± 360 degrees of rotation, and mirroring across the x or y axis. This was done randomly for each image prior to training. Validation images used during training runs were not augmented, nor were testing images. During training, 100 epochs were used unless otherwise specified, with a learning rate of 0.001. All training was performed using MATLAB R2021b with the deep learning and machine learning toolboxes.

#### 2.5.1. Bayesian-Optimized ShrapML Model Evaluation

Training was performed after Bayesian optimization with the new phantom images. All training was performed on an HP workstation (Hewlett-Packard, Palo Alto, CA, USA) running Windows 10 Pro (Microsoft, Redmond, WA, USA) and an Intel Xeon W-2123 (3.6 GHz, 4 core, Santa Clara, CA, USA) processor with 64 GB RAM using an NVIDIA Quatro p1000 (4GB VRAM, Santa Clara, CA, USA) GPU with an image batch size of 32 images.

#### 2.5.2. Comparison to Conventional Image Classification Models

In order to compare the performance of the Bayesian-optimized ShrapML to that of additional classifiers, a literature review was conducted, considering studies published in the last ten years, to identify 11 classifiers that had previous uses either with ultrasound image datasets or in real-time applications. These classifiers are shown in Table 2, which highlights the differences between the complexities in the algorithms by identifying architectural details and the number of parameters.

These additional 11 classifier algorithms were imported into the MATLAB R2021b deep learning toolbox. Input and output layers were adjusted to meet the 512 × 512 image input size and the 2 class types (baseline and shrapnel), respectively. The processed datasets (see Section 2.4) were used for transfer learning with these pretrained models. To identify the highest-accuracy candidates from this original group of 12 models (ShrapML plus 11 conventional classifiers), the models were trained in short runs using 5 epochs, a training image subset of 200 images, and a batch size of 10. Training was performed using CPU on an HP workstation (Hewlett-Packard, Palo Alto, CA, USA) running Windows 10 Pro (Microsoft, Redmond, WA, USA) and an Intel Xeon W-2123 (3.6 GHz, 4 core, Santa Clara, CA, USA) processor with 64 GB RAM. After transfer learning occurred on these new models, an isolated dataset was used to test the model and quantify the performance metrics.

#### 2.5.3. Robust Model Evaluation

After initial model evaluation, the performance metrics were gauged for highest accuracy and lowest training time, which generated a ranking for each model. The top 5 models were selected for robust training. All subsequent training was performed on an HP workstation (Hewlett-Packard, Palo Alto, CA, USA) running Windows 10 Pro (Microsoft, Redmond, WA, USA) and an AMD Ryzen 5 3600X (3.8 GHz, 6 core, Santa Clara, CA, USA) processor with 32 GB RAM using a NVIDIA GeForce RTX 2060 Super (8 GB VRAM, Santa Clara, CA, USA) GPU with an image batch size of 16 images. The same datasets as those used in the initial evaluation were used again for transfer learning and testing.

### 2.6. Performance Metrics

Test-set predictions were performed for each trained model using 20% of the full dataset, which was reserved prior to training. A table of ground truth labels, class predictions, and confidences was generated with all the predictions, which was used for backend analysis of the model’s performance. Confusion matrices were constructed to distinguish true-positive, false-positive, true-negative, and false-negative results. Accuracy, precision, recall, specificity, and F1 score were calculated for each model. ROC curves and the area under the ROC curve (AUC) were also generated. Analyses were performed using MATLAB R2021b, and confusion matrix graphics were created using GraphPad Prism 9 (San Diego, CA, USA). For comparison to other classifier models, the training and testing time was measured for each model to assess relative speeds.

## 3. Results

### 3.1. Bayesian Optimization of ShrapML

We iteratively and selectively developed ShrapML for use in ultrasound image classification in previous studies; however, it was not thoroughly optimized. To this end, 250 Bayesian optimization iterations were performed across six key hyperparameters shown in Table 1. The three top-performing models from this exercise, along with other representative models, are shown in Table 3. For comparison, the selected performance and hyperparameters are also shown from the original ShrapML model. Overall, Bayesian optimization was able to identify higher-performing models than the original ShrapML.

Performance was evaluated in three ways: validation accuracy, validation loss, and training time. Training time was considered a factor of computational burden of the model for potential real-time or near-real-time deployment in real-world applications. In general, a larger fully connected layer at the end of the model and more CNN layers resulted in better performance. The highest-performing model reached 50% lower loss than the original ShrapML, although that performance improvement was paired with four times the training time compared to iterations 83 and 71. As a result, iteration 83 was selected as the optimal version, as it achieved similar validation performance with a much quicker operation vs. iteration 33. The exact architecture of the optimized hyperparameters within ShrapML is shown in Figure 2.

Next, the optimized network was trained with the entire dataset, which included more than 13,000 images and 100 training epochs, in order to refine the model weights. The false-positive rate was slightly higher than the false-negative rate, but both rates remained low (Figure 3A). The ROC curve for the optimized model is shown in Figure 3B. Backend testing resulted in 97% accuracy, with an F1 score and AUC of 0.9765 and 0.9985, respectively (Table 4). Compared to the original ShrapML results [11], accuracy, F1, and AUC were 95%, 0.95, and 0.95, respectively. These represent only slight improvements, although the optimized model was trained on a much larger phantom image set with 10× the number of images when compared to the original ShrapML model results.

### 3.2. Comparison to Others Models

Next, we evaluated how ShrapML compared to conventional image classifiers that have been extensively evaluated using ImageNet. Initially, 11 models were selected, which were, along with ShrapML, trained for five epochs with a reduced dataset of 200 images as an initial streamlined comparison (Table 5). The models were evaluated based on two key performance metrics: test prediction accuracy and training time. The best-performing models based on prediction accuracy were ResNet101 and VGG16, which reached 0.83 accuracy. Others, including ShrapML, surpassed 0.75 in five epochs. Although VGG16 achieved the highest accuracy, it took over an hour to complete training, whereas other models, such as SqueezeNet and ShrapML, took less than 5 min. For this reason, both test accuracy and training time were used to down select, with five models retained for further training: DarkNet19, GoogleNet, MobileNetv2, ShrapML, and SqueezeNet.

Performance comparison of the five down-selected models consisted of more robust training using 100 epochs and the full image dataset (13.3k images). Confusion matrices were compared, and all models showed high true-positive and true-negative rates, with MobileNetv2 having the lowest overall false-positive and false-negative rates of the five models (Figure 4A–E). MobileNetv2 was the best-performing model based on the traditional performance metrics, such as accuracy and F1 score, whereas SqueezeNet had the worst performance (Table 6). However, the difference between Mobile and ShrapML for accuracy and F1 score was 0.032 (Mobile, 0.998 vs. ShrapML, 0.966) and 0.031 (Mobile 0.999 vs. ShrapML 0.967), respectively. This is a minor difference in contrast to inference time, where ShrapML processed testing images in 10.2 milliseconds (ms), whereas Mobile required 104 ms—a 10× difference. In conclusion, ShrapML strikes a balance between standard performance metrics and performance speed, which may be optimal in certain real-time imaging applications.

## 4. Discussion

Ultrasound imaging has a growing value in medical diagnostics, especially when a quick, accurate assessment is needed. Scenarios such as this often happen in emergency medicine and combat casualty care. Trained personnel are required for image interpretation and are commonly not readily available in remote environments. By lowering the cognitive burden and developing automated detection of shrapnel, medical imaging becomes accessible in these extreme environments. Here, we describe the Bayesian optimization of the existing ShrapML classifier algorithm and its comparison to other conventional classifiers trained using the ImageNet archive.

Identification of an algorithm with high accuracy and lower computational power needs will enable integration into various US hardware units for use in such austere environments. As US instrument size shrinks, in some cases to the size of a cellular phone, it becomes ever more critical that this minimal computational power threshold is maintained.

Bayesian optimization of ShrapML improved accuracy to 97%. Conventional classifier algorithms were used to evaluate the performance of transfer learning with ultrasound image sets. These conventional algorithms had millions more parameters and were expected to outperform the smaller ShrapML in terms of accuracy in detecting features because of their additional complexity and size. This was not the case. ShrapML’s accuracy was proven to rival that of the other models, completing training and testing in a fraction of the time compared to conventional algorithms. However, MobileNetv2 and other models with millions of trainable parameters can result in better performance if the highest possible accuracy and F1 score are essential. For triage applications, such as those required in emergency medicine, the tradeoff between speed and accuracy may tip further in the speed direction when compared to AI-focused assistance with a surgical operation [33,34] or identification of a tumor’s precise tissue boundary [35].

There are some limitations of the current work and scope. First, more diversity in the phantom design may be needed to further reduce the possibility of overfitting. This problem can be addressed with future testing or with more robust data augmentation, such as mixup [36,37,38]. Second, the phantom is limited in its complexity when compared to real tissue. Although it may be complex in terms of ultrasound properties, it lacks tissue or-organ-level organization, as well as vessels with pulsatile flow. Next steps should consider transfer learning of the optimal models with animal datasets to improve the training complexity with a more relevant dataset. Third, only shrapnel detection was evaluated in the present study. This was selected as a simple initial use case with a high triage need in military applications; however, more widely used is the extended Focus Assessment with Sonography for Trauma (eFAST) examination procedure for detection of pneumothorax or abdominal hemorrhage. With the identification of an optimal classifier model, shrapnel detection algorithms can eventually be used in eFAST applications, as the same models and principles showcased here can apply.

Next steps for this work will involve transitioning this work into real-time use cases paired with ultrasound imaging. Streamed ultrasound video footage will need to be evaluated to determine whether the compression requirements for video streaming impact model performance. Integration of AI models with tablets or small microcontrollers will also be essential when moving to real time to eliminate the need for a large computer in military or remote medicine situations. Further next steps will look at modifications of this model for use in object detection to precisely locate the foreign body placement instead of only classification. These next steps, along with models optimized for speed and ultrasound imaging, will help to reduce the cognitive load of image interpretation during high-stress, emergency medicine situations.

## 5. Conclusions

In conclusion, artificial intelligence has the potential to improve medical imaging with an appropriate model for a given application. For ultrasound imaging in military and austere environments where resources are limited and high-level triage is the primary goal, simple deep learning models with rapid inference times may be ideal for real-time deployment. The ShrapML algorithm, which we further optimized in this work, is suited for the specific task of rapidly identifying shrapnel much faster than conventional deep learning models. This model will be integrated for use in real time going forward and transitioned to additional ultrasound imaging applications to further highlight the utility that AI can offer for medical imaging applications.

## Figures and Tables

**Figure 1 jimaging-08-00140-f001:**
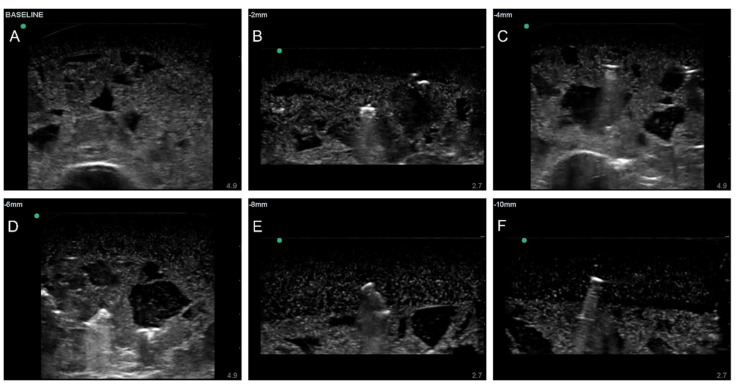
Representative ultrasound images for preprocessed baseline (**A**) and shrapnel of varying sizes—2 mm (**B**), 4 mm (**C**), 6 mm (**D**), 8 mm (**E**), and 10 mm (**F**)—acquired in the gelatin phantom.

**Figure 2 jimaging-08-00140-f002:**
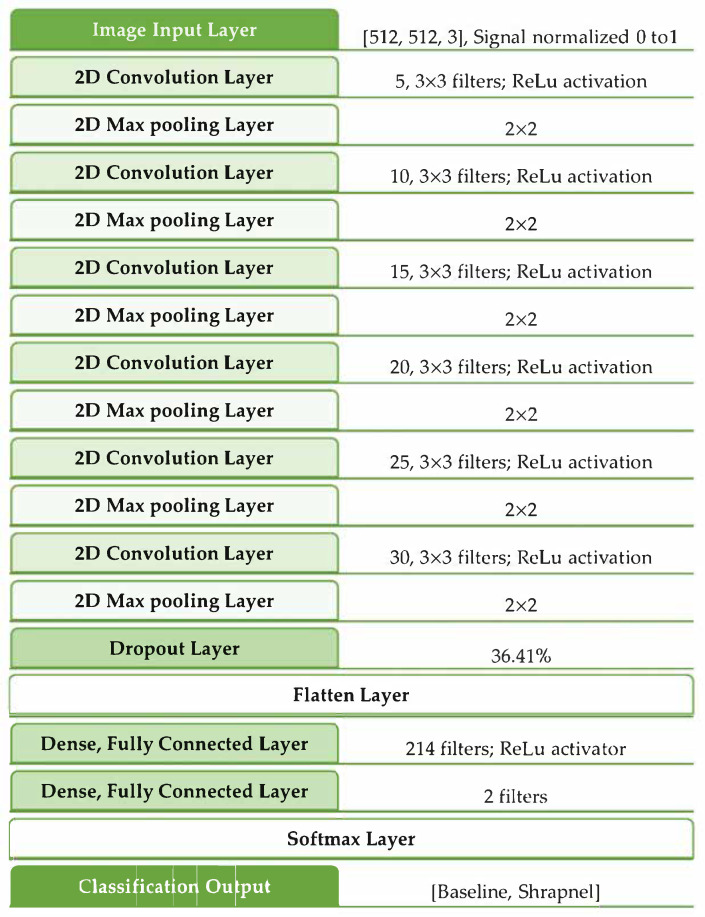
Network architecture for the optimized ShrapML model.

**Figure 3 jimaging-08-00140-f003:**
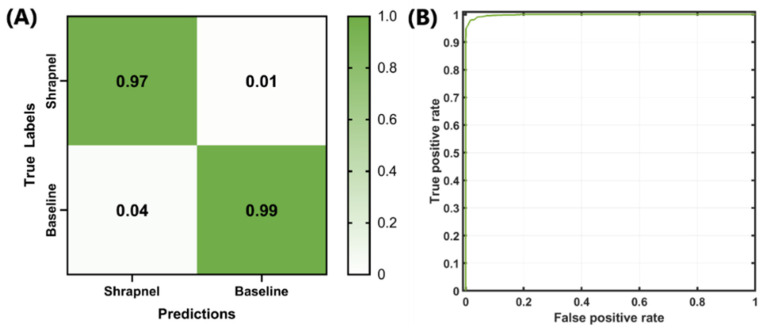
Final backend analysis of trained Bayesian-optimized ShrapML model includes (**A**) confusion matrix, (**B**) ROC analysis from test image sets.

**Figure 4 jimaging-08-00140-f004:**
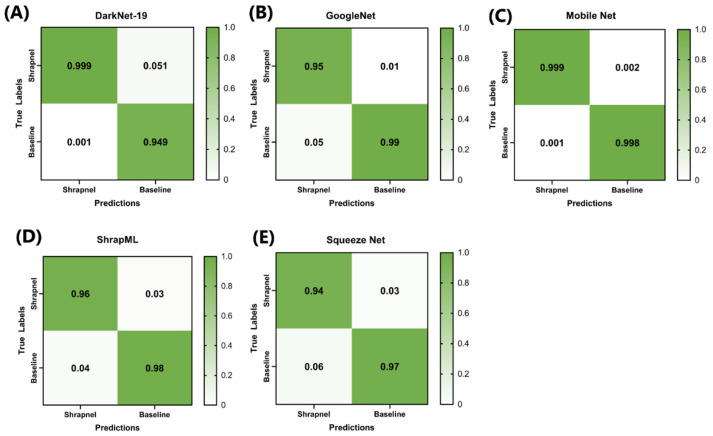
Confusion matrix analysis after 100 training epochs for (**A**) DarkNet-19, (**B**) GoogleNet, (**C**) MobileNetv2, (**D**) ShrapML, and (**E**) SqueezeNet.

**Table 1 jimaging-08-00140-t001:** Summary of Bayesian-optimized hyperparameters for ShrapML.

Hyperparameter	Range of Values	Type
Number of CNN layers	1–6	Integer only
CNN filters	4–32	Integer only
Dropout rate	25–75%	Real number
Fully connected layer filters	8–256	Integer only
Solver type	RMSprop, ADAM, SGDM	Categorical

**Table 2 jimaging-08-00140-t002:** Summary of image classifier model architecture for shrapnel identification in ultrasound image datasets. The algorithm architecture details identify the layer counts as layers with weights: Conv layers and fully connected (FC) or dense layers. This column also contains information about various modules in use (Fire, MBConv, Bottleneck, etc.).

Classifier Algorithm	Architecture Details	Source of Training Images	Parameters (in Millions)	Year First Published
ShrapML	8 layers—6 Conv, 2 FC	Ultrasound datasets	0.43	2022 [11]
AlexNet	8 layers—5 Conv, 3 FC	ImageNet	62.3	2012 [23]
DarkNet19	19 layers—19 Conv	ImageNet	20.8	2016 [24]
DarkNet53	53 layers—52 Conv, 1 FC	ImageNet	41.6	2018 [25]
EfficientNetB0	82 layers—1 Conv, 16 MBConv modules	ImageNet	5.3	2020 [26]
GoogleNet	22 layers—22 Conv	ImageNet	7	2014 [27]
InceptionNetv3	101 layers—99 Conv, 2 FC	ImageNet	23.9	2015 [28]
MobileNetv2	53 layers—3 Conv, 7 Bottleneck modules	ImageNet	3.5	2019 [29]
ResNet50	50 layers—50 Conv	ImageNet	25.6	2015 [30]
ResNet101	101 layers—101 Conv	ImageNet	44.6	2015 [30]
SqueezeNet	18 layers—2 Conv, 8 Fire modules	ImageNet	1.24	2016 [31]
VGG16	16 layers—13 Conv, 3 FC	ImageNet	138	2014 [32]

**Table 3 jimaging-08-00140-t003:** Summary of results of the Bayesian optimization of ShrapML. Iterations 33, 83, and 71 were the three highest-performing iterations, based on validation loss. Iterations 231 and 74 were representative medium- and poor-performing iterations. The last column was results for 10 epochs of training from the original ShrapML model. The single-color heat map indicates best-performing models in green, with the worst being uncolored for each performance metric row.

Model Feature	Iteration 33	Iteration 83	Iteration 71	Iteration 231	Iteration 74	Original ShrapML
FC Nodes	252	214	250	57	8	256
CNN Nodes	32	5	5	23	32	16
Dropout Rate	31.2%	36.4%	25.6%	72.8%	59.8%	55.0%
Solver	ADAM	RMSprop	RMSprop	SGDM	ADAM	RMSprop
# Layers	6	6	6	4	3	5
Time to 10 Epochs	40:52	09:21	09:20	28:53	38:45	24:36
Validation Accuracy	93.7%	93.4%	93.7%	77.8%	54.4%	87.7%
Validation Loss	0.1753	0.2056	0.2296	0.4815	0.6898	0.3448

**Table 4 jimaging-08-00140-t004:** Summary of performance metrics for ShrapML.

Accuracy	Area Under ROC	F1 Score	Precision	Recall	Specificity
0.9761	0.9985	0.9765	0.9645	0.9889	0.9631

**Table 5 jimaging-08-00140-t005:** Performance values of accuracy obtained during testing and time needed to train the 12 models from initial experimental training using five epochs. The five selected models are indicated in bold.

Model	Test Accuracy	Training Time (min)
AlexNet	0.50	10.6
**DarkNet19**	**0.** **79**	**15.6**
DarkNet53	0.68	36.4
EfficientNetB0	0.81	29.5
**GoogleNet**	**0.** **73**	**12.2**
InceptionNetV3	0.58	22.0
**MobileNetV2**	**0.76**	**20.8**
ResNet50	0.75	26.6
ResNet101	0.83	41.1
**ShrapML**	**0.** **77**	**1.3**
**SqueezeNet**	**0.50**	**4.6**
VGG 16	0.83	67.1

**Table 6 jimaging-08-00140-t006:** Summary of performance metrics for each of five models trained using 100 epochs. The single-color heat map indicates the best-performing models in green, with worst being uncolored for each performance metric row.

Metric	DarkNet-19	GoogleNet	Mobile Netv2	ShrapML	SqueezeNet
Accuracy	0.973	0.971	0.998	0.966	0.955
AUC	0.998	0.997	1.000	0.996	0.993
F1	0.973	0.972	0.998	0.967	0.956
Precision	0.999	0.953	0.999	0.958	0.943
Recall	0.947	0.992	0.998	0.976	0.969
Specificity	0.999	0.950	0.998	0.956	0.941
Testing Image Inference Time (ms)	121.40	68.80	104.00	10.20	21.90

## Data Availability

The datasets generated and/or analyzed during the current study are available from the corresponding author upon reasonable request.

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
