# Peer review of "Comparison of Ultrasound Image Classifier Deep Learning Algorithms for Shrapnel Detection"

_2313-433X, 2022, doi:10.3390/jimaging8050140_

Round 1
Reviewer 1 Report
The authors evaluated various deep learning algorithms for shrapnel detection. Overall paper is good. I have the following minor comments:
1) Abstract is missing how the existing DL-based methods are lacking
2) The introduction is too short, I would suggest authors make the background (introduce the problem, the need of research, other recent methods)
3) The dataset is not elaborated thoroughly, did you use a public dataset? is there any need for ethical approval (in the case of the private dataset), In the case of a public dataset provide the links and proper references of the dataset.
4) what is the inference time (per image) for these models?
5) There is no conclusion? I would recommend separating the conclusion from the discussion and including the summarized conclusive outcome of the study
Author Response
Dear Ms. Charlene Cheng,
We would like to thank you and the Reviewers for the thorough review of our manuscript and for giving us the opportunity to respond to the comments. We have reproduced the Reviewer comments below in italics, followed by our responses. Changes to the manuscript are tracked for the Reviewers’ convenience and are reproduced in our responses below when possible. We hope that our responses address all the Reviewers’ concerns and that our study is now suitable for publication.
Sincerely,
Emily N. Boice
Reviewer A
The authors evaluated various deep learning algorithms for shrapnel detection. Overall paper is good. I have the following minor comments:
- Abstract is missing how the existing DL-based methods are lacking
We don’t necessarily believe existing DL methods are lacking, but this paper aims to optimize a network explicitly for shrapnel detection and compare it performance in terms of conventional metrics and overall processing speed. We highlight in the abstract the goal was for getting deep learning algorithms working in real time and thus compare ShrapML vs. conventional models.
- The introduction is too short, I would suggest authors make the background (introduce the problem, the need of research, other recent methods)
We appreciate the observation to our succinct introduction. To keep the overall story clear and brief for the reader, we introduced the problem of how difficult ultrasound image acquisition and interpretation is in the first paragraph. The need for algorithms to simplify this process and several that have been used previously for other medical applications is identified in the second paragraph to showcase their versatility. The recent work from our lab outlining a previously developed algorithm with ultrasound image datasets for the shrapnel detection in the third paragraph is meant to bring the reader context to the scope of this current work. The final paragraph describes additional existing methods using conventional algorithms trained on traditional image datasets (and therefore not trained on ultrasound images). To fairly compare the algorithms, the models would require transfer learning on our data sets and optimization before comparison for our current work with our goal to arrive at an algorithm that is optimized for use in real-time applications. For these reasons, we believe this section of the manuscript properly introduces the study.
- The dataset is not elaborated thoroughly, did you use a public dataset? is there any need for ethical approval (in the case of the private dataset), In the case of a public dataset provide the links and proper references of the dataset.
The data set was collected for this study using gelatin tissue phantoms, so there are no ethical approval requirements in terms of human or live animal image collection. Public databases of shrapnel images were not identified for this work which necessitated the private image repository. The details are discussed in sections 2.3 and 2.4.
- What is the inference time (per image) for these models?
We have added a per image processing time to the results in Figure 4 for the various models as suggested as opposed to a single time for the entire test image sets. This is reflected in Figure 4 and the words referencing those results.
- There is no conclusion? I would recommend separating the conclusion from the discussion and including the summarized conclusive outcome of the study
Thanks for the suggestion, we have added a conclusions section to the paper to more explicitly highlight the primary takeaway points for the manuscript.
Reviewer 2 Report
The work presents the comparison of ultrasound (US) image classifier deep learning-based algorithms for the detection of shrapnel. In doing so, the authors evaluate their previously developed method, which is called ShrapML, against several existing deep learning-based approaches, like AlexNet, GoogleNet, MobileNetv2, ResNet50, etc. The main finding is that while MobileNetv2 reached a higher accuracy than ShrapML there was a trade-off between accuracy and speed, with ShrapML being 10x faster than MobileNetv2. Hence, the next targeted steps for this work will be the transition of the approach into real time use cases paired with ultrasound imaging. Therefore, this work can be seen as an intermediate work, evaluating existing approaches, but having the next step already in mind. In summary, the submission looks like a valid work and the results seem to be reasonable.
Data Availability Statement:
The datasets generated during and/or analyzed during the current study are available from the corresponding author upon reasonable request.
Would be nice (not a must, just a suggestion), if the authors could make their dataset available for the community, e.g. on Figshare (https://figshare.com/) in combination with a data article, like Scientific Data from Nature Publishing Group (NPG):
https://www.nature.com/sdata/
Author Response
Dear Ms. Charlene Cheng,
We would like to thank you and the Reviewers for the thorough review of our manuscript and for giving us the opportunity to respond to the comments. We have reproduced the Reviewer comments below in italics, followed by our responses. Changes to the manuscript are tracked for the Reviewers’ convenience and are reproduced in our responses below when possible. We hope that our responses address all the Reviewers’ concerns and that our study is now suitable for publication.
Sincerely,
Emily N. Boice
Reviewer B
In summary, the submission looks like a valid work and the results seem to be reasonable.
- The datasets generated during and/or analyzed during the current study are available from the corresponding author upon reasonable request. Would be nice (not a must, just a suggestion), if the authors could make their dataset available for the community, e.g. on Figshare (https://figshare.com/) in combination with a data article, like Scientific Data from Nature Publishing Group (NPG).
Thanks for the reviewing the manuscript and the suggestion. We will look into the possibility of sharing the large ultrasound image sets in a public repository, but that was not planned at this time.
Reviewer 3 Report
The paper entitled Comparison of ultrasound image classifier deep learning algorithms for shrapnel detection by Boice et al, developed a deep learning neural-network model expanding the work to further optimize the model and compare its performance with 4 conventional models. The model’s parameters were further refined resulting in an F1 score of 0.98.
The conceptualization of the paper is well but the paper should be restructured.
Please add the section Related work.
The Conclusion section is missed, please add it.
In the section „ShrapML Architecture” the authors did not specify if the images were resized.
All technical methods, hardware, and software are made available to the reader.
The results provided by the proposed model are compared with 11 CNN architectures published between the years 2014 and 2022, these used images that belong ImageNet.
Performance values of accuracy obtained during testing and time were trained on the 12 models from initial experimental training using 5 epochs, five were selected models in function by accuracy and training time.
The study is based on 38 references, of these only four are before 2015.
Author Response
Dear Ms. Charlene Cheng,
We would like to thank you and the Reviewers for the thorough review of our manuscript and for giving us the opportunity to respond to the comments. We have reproduced the Reviewer comments below in italics, followed by our responses. Changes to the manuscript are tracked for the Reviewers’ convenience and are reproduced in our responses below when possible. We hope that our responses address all the Reviewers’ concerns and that our study is now suitable for publication.
Sincerely,
Emily N. Boice
Reviewer C
- Please add the section Related work.
We appreciate this comment and that it highlights the need for publications to contain a section showcasing how unique this method of comparing algorithms is with regards to overall accuracy and speed of image testing. In an effort to keep the story clear and succinct, we incorporated these ideas within the final paragraph of the introduction and in a separate methods table (Table 2). The final paragraph describes widely published conventionally trained algorithms and why algorithms trained using ImageNet datasets would need transfer learning for fair comparison. To place our unique work with phantom ultrasound image datasets within the scope of the field, we then highlight the need for highly accurate algorithms that are also computationally less extensive for eventual incorporation with ultrasound hardware for use in real-time applications. Table 2 in the methods further summarizes algorithm design details for the related work for this manuscript.
- The Conclusion section is missed, please add it.
Thanks for the suggestion, we have added a conclusions section to the paper to more explicitly highlight the primary takeaway points for the manuscript
- In the section “ShrapML Architecture” the authors did not specify if the images were resized.
Yes, the images were resized. After the images are cropped to remove the text on the image, they are sized to 512 x 512. This is mentioned in the methods section on line 136: “US information included in the exported frames followed by converting to 16 bit and re-sizing to 512 x 512.”